# PATH SELECTION MAKES BERT-FAMILY GOOD GENERATORS

## ABSTRACT

Previous research endeavors have sought to tailor BERT-family for non-autoregressive generation tasks through task-specific fine-tuning, and more recent work has also attempted to evolve them into instruction followers after instruction tuning, but their generation quality, even with more powerful iterative decoding methods, continues to lag behind the competitive autoregressive (AR) models. Furthermore, these progress rely on additional fine-tuning procedures, while the generative potential of the BERT-family without fine-tuning still remains unexplored. Hence, we delve deeper into the key issues leading to the performance gaps of BERT-family in generation tasks and put forth innovative solutions. Specifically, existing studies often overlook the significance of the training sequence decomposition format. Unlike autoregressive models that naturally decompose text sequences in a left-to-right fashion during both training and inference, BERT-family are trained using a random decomposition approach (i.e., randomly masking the texts) but strive to identify an optimal composition (referred to decoding paths) during inference. To alleviate this mismatching, we introduce a *path selection* method to expand the search space during inference, facilitating the discovery of more suitable compositions. Additionally, we propose *path selection\**, which further integrates path selection into the training process, enabling the model to learn preferences for specific paths. Our experiments across a range of zero-shot common sense reasoning and reading comprehension tasks and several task-specific generation tasks showcase the substantial performance enhancements for the BERT-family using the proposed methods, reaching the performance levels that are on par with, and in some cases surpassing the AR pre-trained models. Our models and code will be publicly accessible on GitHub.

## 1 INTRODUCTION

In recent years, the progress of large language models, such as Llama (Touvron et al., 2023a;b), Gemini (Team et al., 2023), and GPT-4 (OpenAI, 2023), has revolutionized the natural language processing (NLP) tasks, demonstrating remarkable capabilities across a diverse range of applications. The majority of these models follow a decoder-only autoregressive architecture, drawing inspiration from the successful GPT series models (Radford et al.; 2019; Brown et al., 2020). Conversely, the BERT-family series models (Devlin et al., 2018; Liu et al., 2019; Conneau et al., 2020; He et al., 2020), which have long been recognized for their excellent performance in various language understanding tasks, appear to be progressing at a relatively slower pace recently. Researchers attribute this deceleration to challenges encountered in language generation scenarios, where BERT-family models struggle to achieve competitive performance when compared to autoregressive (AR) models.

Previous works have delved into strategies for leveraging the BERT-family for language generation tasks, particularly following the introduction of the Mask-Predict algorithm (Ghazvininejad et al., 2019), which offers a promising approach for adapting BERT-family to non-autoregressive (NAR) generation scenarios. Researchers have attempted to customize BERT-family for simple specific generation tasks (Chan & Fan, 2019; Jiang et al., 2021; Su et al., 2021; Liang et al., 2023b;a), and recent advancements have also explored the potential for BERT-family in becoming instruction followers after instruction tuning (Xiao et al., 2024). Despite their efforts, performance gaps persist when compared to AR models. Furthermore, these endeavors need additional fine-tuning proce-

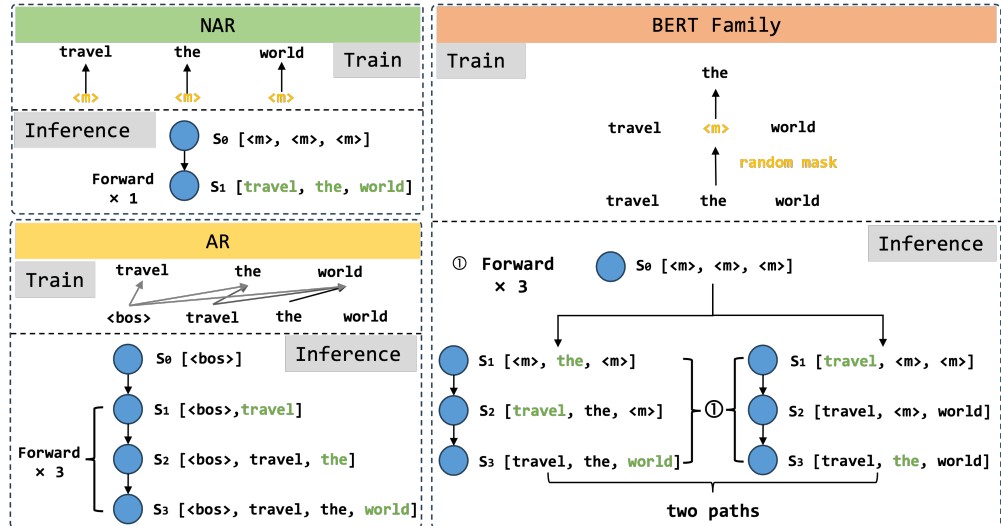

Figure 1: The sequence decomposition for training, and composition methods (i.e., decoding path to achieve the outputs sequence) for different language models.

dures to endow BERT-family models with generation capabilities, thereby leaving the BERT-family without fine-tuning under-explored.

In this paper, we delve deeper into the key issues leading to the performance gaps of BERT-family in generation tasks and put forth innovative solutions. One of the central challenges may stem from the sequence decomposition format. As shown in the Figure 1, AR models naturally decompose language modeling as the task of next-token prediction in a left-to-right order and leverage the corresponding composition strategy during inference. In contrast, BERT-family adopt a random decomposition approach during training (i.e., randomly masking several tokens) but follow certain given criteria to identify an optimal composition (denoted as decoding path in this paper) to achieve target sequences during inference, resulting in a significant training-inference gap. Regarding this mismatching, we propose *path selection* which enables BERT-family models to choose a more appropriate composition from multiple candidates during inference. Additionally, we present *path selection\**, a novel approach that incorporates path selection into the training process, which aims to instruct BERT-family on learning preferences for the outputs achieved by various compositions.

In our study, we introduce a newly developed variant of the BERT-family, which we refer to as **Ge**nerative **BERT (GeBERT)**, to launch a fair comparison with AR models. GeBERT is built on an encoder-only architecture with a bidirectional attention mechanism akin to the conventional BERT-family. Additionally, in line with several current mainstream AR pre-training models (Zhang et al., 2022; Black et al., 2022; Biderman et al., 2023; Peng et al., 2023), we integrate several innovative techniques and adopt a more widely-used training corpus to pre-train GeBERT. Initially, we assess our proposed methods across a range of zero-shot common sense reasoning and reading comprehension tasks using GeBERT without fine-tuning. Our results illustrate that our methods lead to significant performance improvements compared to the baseline GeBERT model, surpassing all AR baselines with comparable model parameters. Besides, we evaluate our methods on two traditional language generation tasks after fine-tuning the models. The corresponding results demonstrate the effectiveness of our proposed methods, with GeBERT achieving state-of-the-art performance in various testing scenarios compared to previous AR and NAR pre-training models.

## 2 PRELIMINARY

### 2.1 UTILIZING BERT-FAMILY FOR LANGUAGE GENERATION TASKS

Previous works (Dong et al., 2019; Wang & Cho, 2019) have theoretically indicated that the BERT-family can be utilized for generating texts by predicting the masked positions in the target sequence.

Despite early efforts by researchers to leverage BERT-family for language generation tasks (Chan & Fan, 2019; Jiang et al., 2021; Su et al., 2021), these attempts yielded suboptimal results compared to the mainstream generative models. Subsequently, researchers attempt to adapt BERT-family to NAR scenarios (Liang et al., 2023b;a; Xiao et al., 2024) via the the Mask-Predict decoding algorithm (Ghazvininejad et al., 2019), which first predicts the entire masked target sequence in the first decoding step, and then refines the target sequence by replacing the unreliable parts with masked tokens and re-generating them in parallel in the subsequent decoding step as details shown in the Appendix A.1, and receives relatively positive feedback regarding performance. During training, these models learn to predict the masked parts in the target sequence, whose loss can be computed as $\mathcal{L} = -\sum_{y_i \in Y_{mask}} \log \mathcal{P}(y_i | Y_{obs}, X; \theta)$, where $X$ denotes the source sequence, $Y_{mask}$ and $Y_{obs}$ are the masked and unmasked parts in the target sequence $Y$, respectively. In this paper, we further delve into the essential technological advancements of BERT-family that leverage the Mask-Predict decoding algorithm to achieve better performance in generation tasks.

## 2.2 DECODING PATHS FOR BERT-FAMILY

Formally, we consider the process of generating a sequence of discrete tokens $Y = (y_1, ..., y_N)$, where $y_i \in V$, a finite vocabulary specific to a language model. This generation process can be interpreted as deterministically sampling a series of successive state spaces $S$, where each state $s_i \in S$ corresponds to a sequence of tokens sampled from $V$, and relies on a policy $\pi$ to transition to the next state. A policy $\pi$ serves as a determinate mapping from states to actions, outlining how the model processes the current sequence and achieves the subsequent sequence in the next state. We denote this specific process to compose the target sequence as the decoding path $P$ of a given language model, where each node represents the current state $s_i$ in $i$th decoding step and each edge represents the policy $\pi_i$ indicating the actions for transitioning from state $s_i$ to $s_{i+1}$.

As shown in Figure 1, different language models have their specific decoding paths to compose the target sequence. The traditional AR and NAR language models typically have a single decoding path for composing a specific target sequence, while BERT-family can explore multiple optional decoding paths, resulting in varied output sequences of differing generation qualities. Selecting a specific decoding path from the multitude of optional paths available in BERT-family is crucial for achieving high-quality outputs. With approximately $2^{TN}$ possible paths for a BERT-family model, as detailed in the Appendix A.2, determining the optimal path is essential for the success of these models. In Ghazvininejad et al. (2019) where the Mask-Predict decoding algorithm was first proposed, the authors heuristically regulate the policy $\pi_t$ in $t$th decoding step as predicting the masked parts in current $Y$ and selecting the specific $n_t$ tokens which are with lowest prediction probabilities to be re-masked, where the number of re-masked tokens can be computed as $n_t = (1 - t/T) * N$, $N$ denotes the total number of tokens in $Y$, $t$ and $T$ denote the current and total decoding step, respectively. While the Mask-Predict algorithm provides a heuristic approach to selecting decoding paths, it may not always yield optimal results. There exist other decoding paths in the candidate space leading to better composition of target sequences (Kreutzer et al., 2020). Hence, we aim to identify an optimal decoding path from such multitudinous candidates by introducing ***path selection*** method. Moreover, we further propose ***path selection\**** which empowers the model to learn the preference between different decoding paths. Our methods seek to enhance the BERT-family's ability to navigate through the complex decoding spaces and generate higher-quality output sequences.

## 3 APPROACH

### 3.1 PATHS SELECTION

We first sample several optional decoding paths from the candidate spaces and select the best one with the highest total prediction probability. Specifically, we follow most of the practice in the Mask-Predict algorithm, except for the selection of the re-masked tokens in each decoding step. As shown in the left of Figure 2, rather than just selecting a specific number of tokens with the lowest prediction probabilities to transform to the unique next state (i.e., the first beam), we allow total $k$ candidate selections for re-masked tokens with the lowest-$k$ total prediction probabilities for each decoding path, where $k$ is the position beam number set in advance. Notice we keep the number of candidate states in each decoding step as $k$, which is similar to the beam search algorithm for AR models (Meister et al., 2020). However, the search times to select those with

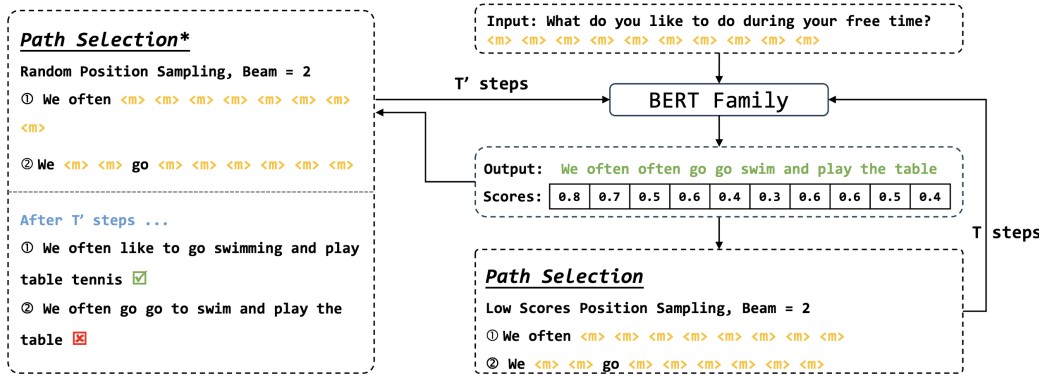

Figure 2: Overview of the path selection and path selection* methods. As for the path selection method during inference, we select the positions for masked tokens with the lowest-$k$ prediction probabilities, while the path selection* randomly samples the positions for masked tokens.

the lowest-$k$ total prediction probabilities is quite large especially when $N$ is large, i.e., given the total decoding step $T$, generated target tokens $N$, and the position beam number $k$, the total search times is $k * \sum_{t \in \{1,2,...,T\}} C_{n_t}^N$, where its detailed proof is in Appendix A.3. Therefore, to reduce the search overhead, we further introduce a simplified version that transforms the search times in $t$th decoding step from $C_{n_t}^N$ to $k$ in which only one position in masked parts can be replaced by the one in unmasked parts to obtain the candidate decoding states, thus the upper bound of search times can be reduced to $T * k^2$. For example, as shown in Figure 2, after obtaining the first beam sequence by Mask-Predict algorithm, we can choose one token in its masked parts with the largest prediction probability (i.e., *go*) to replace the one in its unmasked parts with the least prediction probability (i.e., *often*) to obtain the second beam sequence in each decoding step.

## 3.2 PATHS SELECTION*

Motivated by the recent direct preference optimization (DPO) algorithm (Rafailov et al., 2024) which adopts positive-negative pair samples to train human preferences for language models, we aim to teach BERT-family the decoding path preference by training with positive-negative pair samples achieved from composition methods. Specifically, as shown in the right of Figure 2, given a specific instance in which several tokens in the target sequence are replaced with masked tokens, denoted as $Y_{mask}$, we randomly[1] sample two different decoding paths to generate these masked tokens in multiple steps, then achieve two different output sequences, and the specific output tokens of $Y_{mask}$ are denoted as $Y_{out}^1$ and $Y_{out}^2$, the details of the sampling methods are presented in Appendix A.4. Subsequently, we use a score function $\text{Score}(\cdot)$, such as the exact match accuracy with ground truth tokens or the BLEU score (Papineni et al., 2002), to identify the specific positive and negative ones. Once $\text{Score}(Y_{out}^1) > \text{Score}(Y_{out}^2)$, we adopt $Y_{out}^1$ as the positive output $Y_w$ and $Y_{out}^2$ as the negative output $Y_l$, and vice versa. Finally, following the common practice in online DPO algorithm, given the reference model $\pi_{\text{ref}}$ and the policy model $\pi_\theta$, we first use $\pi_{\text{ref}}$ to sample the positive-negative pair samples, then update $\pi_\theta$ with the DPO loss:

$$\mathcal{L}_{\text{DPO}}(\pi_\theta; \pi_{\text{ref}}) = -\log \sigma \left[ \beta \left( \frac{\pi_\theta(Y_w|Y_{obs}, X)}{\pi_{\text{ref}}(Y_w|Y_{obs}, X)} - \frac{\pi_\theta(Y_l|Y_{obs}, X)}{\pi_{\text{ref}}(Y_l|Y_{obs}, X)} \right) \right], \quad (1)$$

where $X$ denotes the source sequence, $Y_{obs}$ denotes the unmasked parts in $Y$, $\sigma$ denotes the sigmoid function, $\beta$ is the hyperparameter controlling the DPO loss, $\pi_\theta(Y_w|Y_{obs}, X) = \sum_{y_i \in Y_w} \mathcal{P}(y_i|Y_{obs}, X; \theta)$, etc. Besides, we add two penalty terms to reduce the failure cases of DPO as mentioned in Pal et al. (2024), i.e., the model reduces the probabilities of positive outputs and meanwhile more significantly reduces the probabilities of negative outputs, then the probability gap between two outputs will be larger, and the DPO loss will be smaller. However, reducing the

---

[1]We randomly sample the number and specific positions of re-masked tokens to transform to the next state in each decoding path rather than that according the rule in the Mask-predict algorithm mentioned above.

probabilities of positive outputs is contrary to our expectations. The penalty terms can be computed as follows:

$$\mathcal{L}_{\text{PEN}}(\pi_\theta; \pi_{\text{ref}}) = \max\left(0, \log \frac{\pi_{\text{ref}}(Y_w|Y_{obs}, X)}{\pi_\theta(Y_w|Y_{obs}, X)}\right) + \max\left(0, \log \frac{\pi_{\text{ref}}(Y_l|Y_{obs}, X)}{\pi_\theta(Y_l|Y_{obs}, X)}\right). \quad (2)$$

Then, combining the above DPO loss and the penalty terms with the traditional masked language modeling loss in BERT-family as mentioned in Section 2.1, which aims to predict the masked tokens:

$$\mathcal{L}_{\text{MLM}}(\pi_\theta) = - \sum_{y_i \in Y_{mask}} \log \mathcal{P}(y_i|Y_{obs}, X; \theta). \quad (3)$$

Our final training loss can be computed as $\mathcal{L} = \mathcal{L}_{\text{MLM}} + \lambda_1 \mathcal{L}_{\text{DPO}} + \lambda_2 \mathcal{L}_{\text{PEN}}$, where $\lambda_1$ and $\lambda_2$ are the hyperparameters to balance the different loss items.

## 4 EXPERIMENTS

### 4.1 IMPLEMENTATION DETAILS

**Backbone Models**  For better evaluation of various generation tasks, we pre-train new variants of BERT-family with a modified masking mechanism during training, which aims to better equip these masked language models for tasks involving generation (Liang et al., 2023b; Xiao et al., 2024), thus we name our model as **Ge**nerative **BERT (GeBERT)**. Specifically, unlike earlier BERT-like models that only mask 15% of tokens in each instance for prediction (Devlin et al., 2018; Liu et al., 2019), we first decompose the training instance into two parts which simulates a scenario akin to conditional generation. Drawing inspiration from prior practice (Song et al., 2019; Li et al., 2022b; Guo et al., 2020; Xiao et al., 2023), we assign different masking methods for these two parts to enable GeBERT to learn both understanding and generation capabilities. Details of our pre-training task are presented in the Appendix A.5. Based on our modified pre-training task, we pre-train two versions of GeBERT containing 124M and 352M parameters which are similar to the base and large versions of other previous pre-trained language models (Devlin et al., 2018; Lewis et al., 2019; Raffel et al., 2020; Huang et al., 2023), denoted as **GeBERT-124M** and **GeBERT-352M**.

**Pre-training Settings**  As for the model architecture, we follow the most practice in previous BERT-like models to build an encoder-only language model with a bi-directional attention mechanism and further incorporate several effective techniques: 1) We use Rotary Positional Embedding (RoPE) (Su et al., 2024) to inject positional information into GeBERT rather than the traditional absolute/relative position encoding. 2) We adopt swiglu Shazeer (2020) as our activation function rather than the traditional ReLU. For pre-training corpus, we adopt the Pile (Gao et al., 2020; Biderman et al., 2022), which is a curated collection of English language datasets containing around 300B tokens and has been widely used for training language models (Zhang et al., 2022; Black et al., 2022; Biderman et al., 2023; Peng et al., 2023). During training, we set the max length as 2048 and pre-train GeBERT for 150k update steps (1 epoch on the Pile) with a batch size of 1024 samples (i.e., 2M tokens). We use Adam optimizer (Kingma & Ba, 2014) with a weight decay of 0.01, the learning rate warms up to 6e-4 in the first 1.5k steps and then decreases gradually with `cosine` decay strategy. We utilize the Megatron-Deepspeed [2] library to train GeBERT on 8 NVIDIA A100-PCIE-80GB GPU cards. We present the details for the model and pre-training in the Appendix A.6.

**Fine-tuning Settings**  We follow the training procedure in previous works (Liang et al., 2023b; Xiao et al., 2024) to fine-tune GeBERT on downstream datasets for non-autoreressive sequence generation tasks. For the fine-tuning settings, we tune the learning rate from {1e-5, 2e-5, 5e-5, 1e-4} for different downstream tasks. We train for a total of 50 epochs and validate the model after each epoch, then obtain the final model with the best validation performance. During the training of the path selection* method, we initialize the policy and reference model with that after fine-tuning for downstream sequence generation tasks. Then, we freeze the parameters of the reference model and only update the parameters of the policy model with the same dataset adopted in fine-tuning. We set the learning rate as 2e-5 and other training hyperparameters the same in the fine-tuning stage. Then,

---

[2]https://github.com/microsoft/Megatron-DeepSpeed

Table 1: Results on zero-shot common sense reasoning and reading comprehension tasks. The first line of GeBERT denotes the baseline which adopts the same decoding path as AR models. **Bold** values denote the best average result (**AVG.**) through all models. underlined values denote the result of our methods outperforming the baseline GeBERT. The abbreviations **Wino.**, **Hella.**, and **Truth.** denote the WinoGrande, Hellaswag, and Truthfulqa datasets, respectively.

| Models | LogiQA | Sciq | ARC-E | ARC-C | Wino. | BoolQ | PIQA | SIQA | Race | Hella. | Truth. | AVG. |
|---|---|---|---|---|---|---|---|---|---|---|---|---|
| *≈ 150M parameters* | | | | | | | | | | | | |
| OPT-125M | 27.93 | 75.2 | 43.52 | 22.78 | 50.28 | 61.07 | 62.02 | 37.21 | 30.05 | 31.25 | 23.99 | 42.31 |
| GPT-neo-125M | 28.88 | 76.5 | 43.73 | 23.12 | 50.43 | 62.02 | 62.46 | 37.21 | 27.56 | 30.40 | 25.83 | 42.56 |
| Pythia-160M | 24.27 | 75.4 | 43.64 | 23.63 | 51.30 | 62.14 | 61.97 | 36.90 | 28.71 | 30.30 | 24.97 | 42.11 |
| RWKV-169M | 24.73 | 75.2 | 47.52 | 23.46 | 50.67 | 62.17 | 64.04 | 37.00 | 26.89 | 32.25 | 22.25 | 42.41 |
| GeBERT-124M | 27.65 | 80.3 | 42.13 | 22.10 | 50.75 | 62.17 | 60.66 | 36.49 | 28.90 | 29.76 | 24.60 | 42.27 |
| + Path Selection | 27.65 | 81.8 | 42.09 | 22.36 | 51.87 | 62.17 | 59.69 | 36.80 | 29.28 | 31.70 | 25.70 | 42.89 |
| + Path Selection* | 28.88 | 80.5 | 42.47 | 22.18 | 52.72 | 62.17 | 60.88 | 36.94 | 29.76 | 32.25 | 25.74 | **43.14** |
| *≈ 350M parameters* | | | | | | | | | | | | |
| OPT-350M | 28.57 | 74.90 | 44.19 | 23.98 | 52.49 | 61.87 | 64.74 | 39.30 | 29.76 | 32.66 | 23.50 | 43.27 |
| Pythia-410M | 29.34 | 81.30 | 52.10 | 24.32 | 53.20 | 61.68 | 67.08 | 38.95 | 30.91 | 40.52 | 23.50 | 45.72 |
| RWKV-430M | 24.42 | 79.00 | 52.23 | 25.17 | 52.80 | 62.05 | 68.44 | 38.84 | 28.71 | 40.78 | 22.28 | 44.98 |
| GeBERT-352M | 28.88 | 83.10 | 51.43 | 23.86 | 52.93 | 62.17 | 65.21 | 39.02 | 30.68 | 40.12 | 24.35 | 45.01 |
| + Path Selection | 29.87 | 83.60 | 51.65 | 24.24 | 52.87 | 62.17 | 65.03 | 39.26 | 30.83 | 41.03 | 25.58 | 46.03 |
| + Path Selection* | 30.33 | 83.30 | 51.97 | 24.18 | 53.19 | 62.17 | 65.78 | 39.51 | 31.00 | 41.30 | 25.80 | **46.21** |

we train the model with 5 epochs. As for the DPO training of the vanilla GeBERT, we initialize the policy and reference model with the final saved checkpoint during pre-training. We sampled a small subset from the pile to conduct DPO training and avoid introducing extra training data.

## 4.2 EVALUATION DETAILS

**Datasets and Metrics** We evaluate our proposed methods on common downstream task-specific generation tasks, which have been widely used in previous pre-trained AR and NAR works, and various zero-shot common sense reasoning and reading comprehension tasks, which are popular to evaluate the vanilla version of current large language models without fine-tuning (Zeng et al., 2022; Touvron et al., 2023a;b). To the best of our knowledge, we are the first to evaluate the pre-trained NAR models for these zero-shot tasks. Specifically, For downstream task-specific generation tasks, we adopt XSUM (Narayan et al., 2018) for the summarization task and MSQG ( MicroSoft Question Generation) dataset for the question generation task from the GLGE benchmark (Liu et al., 2021). For the evaluation metrics, we adopt ROUGE F1 (ROUGE-1/2/L) (Lin & Hovy, 2002) for XSUM, and BLEU (BLEU-4) (Papineni et al., 2002), Rouge-L and METEOR (Lavie & Agarwal, 2007) for MSQG. For zero-shot common sense reasoning and reading comprehension tasks, we adopt ARC-easy, ARC-challenge (Clark et al., 2018), BoolQ (Clark et al., 2019), PIQA (Bisk et al., 2020), SIQA (Sap et al., 2019), WinoGrande (Sakaguchi et al., 2021), Race (Lai et al., 2017), Sciq (Johannes Welbl, 2017), LogiQA (Liu et al., 2020), Hellaswag (Zellers et al., 2019), and Truthfulqa (Lin et al., 2021), which are all widely used for evaluating recent language models. We adopt Language Model Evaluation (Gao et al., 2021) framework to evaluate these datasets under a zero-shot setting (Biderman et al., 2023). We adopt normalized accuracy for PIQA, ARC-challenge, LogiQA, Hellaswag, and accuracy for other tasks following previous works (Biderman et al., 2023).

**Baseline Models** For the downstream task-specific generation tasks, we adopt the vanilla Transformer baseline (Vaswani et al., 2017) and previous pre-trained AR models including MASS (Song et al., 2019), BART Lewis et al. (2019), and ProphetNet (Qi et al., 2020) which are included in the official GLGE evaluation leaderboard as autoregressive baselines. For NAR baselines, we adopt the previous pre-trained NAR models including BANG (Bang et al., 2023), ELMER (Li et al., 2022a) and PreDAT (Huang et al., 2023). Besides, we also include MIST (Jiang et al., 2021) and DEER (Liang et al., 2023a) which also fine-tune the traditional BERT-family to complete the generation tasks. For common sense reasoning and reading comprehension tasks, which are only widely used after the popularity of large language models and never been included in the evaluation of previous NAR models, we adopt the recent large language models which are also trained on the Pile for around 300B tokens and contains the comparable model parameters with GeBERT, including OPT-125M/350M (Zhang et al., 2022), GPT-neo-125M (Black et al., 2022), Pythia-160M/410M (Bider-

Table 2: Results on task-specific generation tasks. **Bold** denotes the best result. underlined values denote the result of our methods outperforming the baseline GeBERT.

| Model | XSUM | | | MSQG | | | Speedup |
|---|---|---|---|---|---|---|---|
| | Rouge-1 | Rouge-2 | Rouge-L | Rouge-L | BLEU-4 | METEOR | |
| Transformer | 30.66 | 10.80 | 24.24 | 29.43 | 4.61 | 9.86 | - |
| *Base Version ($\approx$ 150M parameters)* | | | | | | | |
| BANG | 32.59 | 8.98 | 27.41 | - | - | - | - |
| ELMER | 37.30 | 13.17 | 29.92 | - | - | - | - |
| PreDAT | 39.79 | 17.38 | 32.71 | - | - | - | - |
| MIST | 34.63 | 11.29 | 28.70 | - | - | - | - |
| DEER | 39.10 | 16.80 | 32.40 | 38.70 | 9.70 | 23.30 | - |
| MASS-base | 39.70 | 17.24 | 31.91 | 38.90 | **10.20** | 23.30 | - |
| BART-base | 38.79 | 16.16 | 30.61 | 38.20 | **10.20** | 22.90 | 1.0x |
| ProphetNet-base | 39.89 | 17.12 | 32.07 | 37.10 | 9.10 | 22.30 | - |
| GeBERT-124M | 40.32 | 16.90 | 32.54 | 39.13 | 9.66 | 23.50 | 3.1x |
| + Path Selection | 40.52 | 17.11 | 32.71 | 39.06 | 9.52 | 23.51 | 1.2x |
| + Path Optimization | 40.92 | **17.39** | **33.08** | 39.46 | 9.72 | **23.68** | 1.2x |
| *Large Version ($\approx$ 350M parameters)* | | | | | | | |
| MASS-middle | 39.10 | 16.50 | 31.40 | 38.90 | 9.50 | 23.50 | - |
| BART-large | **45.10** | **22.20** | **37.20** | 38.80 | 9.20 | 24.30 | - |
| ProphetNet-large | 44.40 | 21.30 | 36.40 | 38.30 | 9.60 | 23.30 | - |
| GeBERT-352M | 44.12 | 21.03 | 36.27 | 39.32 | 10.23 | 23.87 | - |
| + Path Selection | 44.33 | 21.23 | 36.40 | 39.38 | 10.21 | 23.90 | - |
| + Path Optimization | 44.84 | 21.89 | 36.89 | **39.78** | **10.29** | 24.32 | - |

man et al., 2023), and RWKV-169M/430M (Peng et al., 2023). We re-run all the baseline models under the same Language Model Evaluation framework (Gao et al., 2024) using their open-source Hugging Face models to ensure consistent evaluation procedures.

### 4.3 MAIN RESULTS

**Zero-shot common sense reasoning and reading comprehension** We present the results on various zero-shot common sense reasoning and reading comprehension tasks in Table 1. Compared with GeBERT and the previous AR models, we can find that: (1) GeBERT can also complete these zero-shot tasks and achieve comparable performance while adopting the same decoding path during inference[3]. (2) Our final models (i.e., GeBERT with path selection and path selection*) achieve the best performance through all the previous AR models on average, outperforming the previous best models (i.e., GPT-neo-125M and Pythia-410M) by around 0.8 and 0.5 score. (4) GeBERT is better at reading comprehension tasks which enable the model to answer questions given supports or evidences such as Sciq and LogiQA, we attribute this to the bi-directional attention mechanism of GeBERT. Besides, Compared with baseline GeBERT which adopts the same decoding path as AR models and those with our path selection and path selection* methods, we can find that: (1) With the path selection method, GeBERT outperforms the baseline GeBERT in most of the evaluation tasks, leading to 0.6/1.0 performance improvements on GeBERT-124M/352M. (2) Further, with the path selection* method, GeBERT can outperform the baseline GeBERT in 10 of 11 evaluation tasks and be on par in BoolQ, leading to around 1.0 performance improvements on average. (3) By comparing GeBERT only with the path selection method and with both proposed methods, the former can achieve performance improvements on most tasks, indicating the effectiveness of the path selection* method. However, the path selection* may also result in performance declines in several tasks, such as Sciq for GeBERT-124M and ARC-C for GeBERT-352M.

**Task-specific generation** Table 2 presents the results on task-specific generation task. We can find that: (1) For the summarization task, though GeBERT-352M underperforms BART-large GeBERT-124M, it outperforms all the other baseline models in all evaluation metrics, indicating that GeBERT can generate more informative and reasonable summaries. (2) For the question generation task, GeBERT-124M outperforms all the baseline models on Rouge-L and METEOR and only presents

---

[3]For GeBERT, we append a masked token after the current sequence and enable the model to predict it, thus realizing the same decoding path as AR models that adopt the policy as predicting the next token.

Table 3: Results of different methods to select the decoding paths for zero-shot common sense reasoning and reading comprehension tasks. **Hyper.** denotes the corresponding hyperparameter.

| Hyber. | LogiQA | Sciq | ARC-E | ARC-C | Wino. | BoolQ | PIQA | SIQA | Race | Hella. | Truth. | AVG. |
|---|---|---|---|---|---|---|---|---|---|---|---|---|
| *multi-step-based* | | | | | | | | | | | | |
| $T = 1$ | 23.50 | 64.6 | 36.49 | 21.93 | 50.75 | 62.17 | 54.19 | 34.75 | 23.44 | 28.15 | 21.42 | 38.30 |
| $T = 4$ | 26.73 | 80.4 | 41.20 | 21.33 | 50.99 | 62.17 | 57.24 | 36.64 | 28.71 | 30.74 | 25.95 | 42.01 |
| $T = 7$ | 26.42 | 80.5 | 41.04 | 22.19 | 52.41 | 62.17 | 58.16 | 36.54 | 29.67 | 31.11 | 24.48 | 42.24 |
| *multi-token-based* | | | | | | | | | | | | |
| $n_{new} = 2$ | 29.03 | 71.1 | 40.15 | 22.44 | 50.99 | 62.17 | 59.19 | 36.89 | 29.47 | 31.68 | 25.09 | 41.65 |
| $n_{new} = 3$ | 27.96 | 66.8 | 37.79 | 22.36 | 50.12 | 62.17 | 57.07 | 35.31 | 27.94 | 30.83 | 24.97 | 40.30 |
| $n_{new} = 4$ | 29.65 | 65.5 | 38.39 | 22.53 | 49.17 | 62.17 | 55.06 | 35.47 | 27.37 | 30.40 | 24.24 | 40.00 |
| *multi-order-based* | | | | | | | | | | | | |
| left-to-right | 27.65 | 80.3 | 42.13 | 22.10 | 50.75 | 62.17 | 60.66 | 36.49 | 28.90 | 29.26 | 24.60 | 42.26 |
| right-to-left | 28.57 | 72.2 | 30.73 | 22.35 | 49.17 | 58.87 | 55.22 | 33.78 | 25.55 | 30.12 | 25.70 | 39.30 |
| random | 25.81 | 79.6 | 41.71 | 21.67 | 50.20 | 62.17 | 56.09 | 33.93 | 28.13 | 29.50 | 24.48 | 41.20 |
| easy-to-hard | 29.19 | 80.4 | 41.96 | 22.44 | 52.57 | 62.17 | 59.79 | 36.80 | 29.79 | 31.73 | 25.58 | 42.90 |
| hard-to-easy | 26.88 | 80.4 | 41.50 | 21.16 | 50.04 | 62.17 | 58.92 | 36.34 | 26.22 | 31.03 | 24.97 | 41.78 |
| *multi-beam-based* | | | | | | | | | | | | |
| $k = 2$ | 28.11 | 81.4 | 42.59 | 22.01 | 52.01 | 62.17 | 59.69 | 36.54 | 30.05 | 31.98 | 25.45 | 42.94 |
| $k = 3$ | 27.65 | 81.8 | 42.09 | 22.36 | 51.86 | 62.17 | 60.01 | 36.80 | 29.67 | 32.17 | 25.58 | 42.90 |
| $k = 4$ | 28.26 | 81.8 | 42.51 | 22.61 | 51.07 | 62.17 | 60.28 | 36.28 | 29.28 | 32.09 | 25.70 | 42.91 |

performance gaps compared with the best baseline models on BLEU-4. GeBERT-352M achieves the best performance across various models on all evaluation metrics. (3) Compared to the GeBERT baseline, which adopts the original vanilla Mask-Predict algorithm to generate the output sequence, the path selection and path selection* methods can bring performance improvements on the XSUM dataset for both GeBERT-124M/352M, indicating that these two methods can enable the model to achieve better performance in generating relatively long targets. However, the path selection method does not lead to consistent performance improvements on the MSQG dataset, which contains relatively short targets. We attribute this to that short sequences will lead to relatively small candidate space and redundant outputs for different decoding paths, thus we can not achieve better outputs from multiple candidates. (4) We also compare the decoding efficiency of GeBERT-124M and BART-base, which contains around 140M parameters, and the results demonstrate that GeBERT can achieve 3.1x speedup with the vanilla Mask-predict algorithm due to the NAR attribute. Further, with path selection and path selection* methods, which will bring the extra search overhead for different decoding paths, GeBERT still achieves a faster generation process, leading to a 1.2x speedup compared to BART.

## 5 ANALYSIS

### 5.1 DISCUSSION OF DIFFERENT METHODS TO DETERMINE THE DECODING PATHS

In Table 1, we report the performance of GeBERT which adopt the same decoding path with AR models and that with our proposed path selection method. However, there still exist multitudinous optional decoding paths to generate the final target sequences as mentioned in section 3.1, we adopt GeBERT-124M to compare the different methods to determine the decoding paths here with the following rules. Notice in Table 1, we regulate the number of newly generated tokens (denoted as $n_{new}$) in each decoding step as $n_{new} = 1$ to keep consistent with AR models, i.e., we generate only one token in a left-to-right order or with the highest-$k$ prediction probabilities in each decoding step, and adopt the total decoding steps adaptive to the target length. Then, we can (1) set $n_{new} = 2/3/4$, and the corresponding decoding steps as $\lceil N/2 \rceil, \lceil N/3 \rceil, \lceil N/3 \rceil$, where $N$ denotes the total target tokens, we denote this method as multi-token-based, (2) set the total decoding steps as $T = 1/4/10$, and the corresponding $n_{new} = \lceil N/1 \rceil, \lceil N/4 \rceil, \lceil N/10 \rceil$. We denote this method as multi-step-based. Besides, with $n_{new} = 1$, there still exist different rules to achieve the specific generated token. (3) Rather than a left-to-right order in Table 1, we can also include a right-to-left or a random order to achieve the final target sequence. Instead of selecting the token with the highest prediction probability, which denoted as a easy-to-hard order (Kasai et al., 2020), we include a hard-to-easy order which generate the token with the lowest prediction probability first. These method can be

denoted as multi-order-based. (4) We can set different beam number $k$ in our proposed path selection method, we compare $k = 2/3/4$ here, denoted as multi-beam-based.

We present the corresponding results with the above methods in Table 3, we can find that: (1) The performance declines as the $n_{new}$ increases, indicating that setting $n_{new} = 1$ to keep consistent with AR models, in which the model predicting one token in each decoding step, is important to achieve competitive performance. (2) With the multi-step-based method, more decoding steps lead to better performance, which also verifies the above observation, i.e., the length of targets is less than the decoding steps in several tasks, such as Sciq and SIQA, where the model will also predict one token in each decoding step. Conversely, the performance on these tasks which contain the relatively long targets such as PIQA and ARC still falls behind the left-to-right baseline. (3) Based on different orders, the easy-to-hard order which we adopt in the path selection method perform best, while several other orders will lead to significant performance declines such as right-to-left order. (4) Adopting different beams in our path selection method performs differently for various tasks but achieve a comparable score on average, and all outperforms the left-to-right baseline, indicating the effectiveness of the path selection method.

## 5.2 COMPARISON WITH TOKENS-AWARE BEAM SEARCH

The path selection method sampling several position beams to achieve multiple candidate outputs is similar to the token-aware beam search algorithm, which has been widely used in AR models (Meister et al., 2020). The token-aware beam search algorithm selects more candidate tokens during inference rather than always the one with the highest prediction probability, which can significantly improve the performance.

Table 4: Results of different beam search algorithms.

| Method | Rouge-1 | Rouge-2 | Rouge-L |
|---|---|---|---|
| GeMLM | 40.32 | 16.90 | 32.54 |
| w/ Token Beam | 40.17 | 16.88 | 32.50 |
| w/ Position Beam | 40.52 | 17.11 | 32.71 |
| w/ Path Selection* | 40.78 | 17.30 | 33.01 |
| w/ Token Beam | 40.58 | 17.19 | 32.90 |
| w/ Position Beam | 40.92 | 17.39 | 33.08 |

We also extend this into BERT-family to permit more optional tokens in each decoding step. Specifically, we randomly select one token in the unmasked parts in target sequence and replace it with one whose prediction probability is below the first one in the overall probability distribution. Compared with our proposed path selection method, the position beams select candidates with different positions based on specific tokens while the token-aware beam search algorithm selects candidates with different prediction tokens based on specific positions. We adopt GeBERT-124M to conduct analytic experiments on XSUM, where the results are presented in Table 4. We can find that the path selection method can achieve consistent performance improvements, but the token-aware beam search algorithm does not work in this scenario. We attribute the failure of token-aware beam search to the different modeling paradigm of BERT-family compared to AR models.

## 5.3 ABLATION STUDY OF PATH SELECTION*

In this section, we conduct an ablation study to explore the effects on different $\lambda_1$ and $\lambda_2$ in our final training loss as mentioned in Section 3.2. We report the performance of $\lambda_1$ in $\{0.0, 0.1, 0.5, 1\}$, $\lambda_2$ in $\{0, 1, 5, 10\}$ without adopting position beams. Compared with the baseline model (i.e., $\lambda_1 = 0.0$, $\lambda_2 = 0$), we can find that (1) $\mathcal{L}_{DPO}$ and $\mathcal{L}_{PEN}$ are both necessary for performance improvements. With $\lambda_1 = 0.5$ and $\lambda_2 = 0$, the performance even declines, indicating the failure cases as mentioned in Section 3.2. (2) In other cases, the per-

Table 5: Result of different $\lambda_1$ and $\lambda_2$.

| Hyperparameter | Rouge-1 | Rouge-2 | Rouge-L |
|---|---|---|---|
| $\lambda_1 = 0.0, \lambda_2 = 0$ | 40.32 | 16.90 | 32.54 |
| $\lambda_1 = 0.5, \lambda_2 = 0$ | 39.85 | 16.88 | 32.52 |
| $\lambda_1 = 0.5, \lambda_2 = 1$ | 40.76 | 17.25 | 32.96 |
| $\lambda_1 = 0.5, \lambda_2 = 5$ | 40.78 | 17.30 | 33.01 |
| $\lambda_1 = 0.5, \lambda_2 = 10$ | 40.70 | 17.24 | 32.97 |
| $\lambda_1 = 0.0, \lambda_2 = 5$ | 40.22 | 16.90 | 32.52 |
| $\lambda_1 = 0.1, \lambda_2 = 5$ | 40.74 | 17.20 | 32.92 |
| $\lambda_1 = 0.5, \lambda_2 = 5$ | 40.78 | 17.30 | 33.01 |
| $\lambda_1 = 1.0, \lambda_2 = 5$ | 40.78 | 17.28 | 33.05 |

formances are close to each other with only around 0.1 gaps on all metrics, indicating that we need

not spend lots of effort to tune the $\lambda_1$ and $\lambda_2$. Our DPO training objective is easy to achieve the corresponding performance improvements.

## 6 DISCUSSIONS

**Discussion with other non-monotonic generation models.**   While BERT-family adopting the Mask-Predict algorithm can be viewed as a special case for non-monotonic generation, we also present the discussion with other works that also explore the non-monotonic generation for language models (Welleck et al., 2019; Shih et al., 2022; Shen et al., 2023). These models can also generate output sequences beyond a strict left-to-right order but still differ from the BERT-family. Specifically, these models only generate one token via one decoding step, rather than adopting the parallel decoding paradigm allowing generating multiple tokens for BERT-family. Furthermore, BERT-family adopt the masking language modeling during training, differing from other models which adopt the variants of autoregressive modeling.

**Discussion with other path selection methods.**   In the realm of NAR translation, prior research has also delved into enhancing output quality by selecting the optimal result from multiple candidate generations, such as CTC-decoding (Graves et al., 2006; Shu et al., 2020; Shao & Feng, 2022) and DA-Transformer (Huang et al., 2022; Shao et al., 2022a; Huang et al., 2023). Nevertheless, these methods typically adhere to a left-to-right order when sampling various candidates. Conversely, our proposed path selection method eliminates order constraints, facilitating the candidates with more diversity. Besides, while some studies have advocated for training models with multiple references (Shao et al., 2022b; Huang et al., 2022; Liu et al., 2023), these methods treat all various references as equally plausible outputs and train with the same weights across them. In contrast, our path selection* method incorporates positive-negative sample pairs into the training process, enabling the model to learn preferences among different candidates.

**Discussion with other DPO methods for AR models**   The Direct Preference Optimization (DPO) algorithm has found widespread application in AR models (Yang et al., 2023; Rafailov et al., 2024; Pal et al., 2024), facilitating the learning of preferences for various output sequences generated through different sampling methods, which can significantly enhance the overall generation quality. Conversely, few explorations have been conducted for BERT-family. In this paper, we propose path selection* to enable BERT-family to learn the preference between different outputs achieved by different decoding paths according to the generation character of BERT-family. While AR models allow different lengths of sampled positive-negative pairs, we recognize that ensuring a consistent sampling length across different pairs is quite important for our methods for fair comparisons between different decoding paths. Another difference between our methods and those for AR models is that there is no significant quality difference between our sampled pairs, which will easily lead to the failure cases as mentioned in Section 3.2. Consequently, integrating penalty terms to stabilize the DPO training process emerges as a crucial aspect of our methods.

## 7 CONCLUSION

In this paper, we pre-train a new version of BERT-family to explore the potential of BERT-family for good generators. According to the specific generation formats of sequences, we propose path selection and path selection* methods to enhance the generation capabilities of these models. Results on a range of zero-shot common sense reasoning and reading comprehension tasks demonstrate that BERT-family can also achieve state-of-the-art performance by adopting our methods compared to current mainstream AR language models, and the experiments on several task-specific generation tasks further verify the effectiveness of our methods, outperforming the previous pre-trained language models. In overall, our works first pre-train a version of BERT-family fron scratch for fair comparison with current AR pre-trained models, and further verify the very promising techniques to unlock the potential of BERT-family for various generation tasks both without and with fine-tuning. We hope our explorations can expand the applications of BERT-family in AI community. In the future, we will further explore larger versions after observing the scaling ability from 124M to 352M of GeBERT, then adapt GeBERT to more type of tasks, such as code generation and math reasoning.

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

# A APPENDIX

## A.1 DETAILS OF THE MASK-PREDICT ALGORITHM

We present an example adopting the Masked-Predict algorithm to generate the output sequence in Figure 3. Specifically, given the prompt, we first initial the input as fully masked tokens (i.e., Input 1) and send it into the model. After the model predict the outputs (i.e., Output 1), we will select specific unreliable tokens with relatively lower prediction probabilities to mask again (i.e., the yellow parts in outputs). In the subsequent decoding step, the model will predict these masked tokens and select several unreliable tokens again. We obtain the final target sequence until reaching the total number of decoding steps set advance. This decoding algorithm assume that the target sequence will be refined better through multiple decoding steps.

Figure 3: Presentation of the Masked-Predict algorithm.

## A.2 DETAILS OF THE DECODING PATHS FOR BERT-FAMILY

**Lemma 1** *Given the total length $N$ of target sequence $Y$, the total decoding step $T$, the total number of optional decoding paths is around $2^{TN}$, exactly $\sum_{m=0}^{N}(-1)^m C_m^N 2^{(N-m)T}$ with the constraint that all tokens in $Y$ should be predicted.*

**Proof 1** *During each decoding step, we can select any subset of $V_Y$, i.e., the model can generate 1 to $N$ different tokens at different position candidates. There exist $C_0^N + C_1^N + C_2^N + ... + C_N^N = 2^N$ candidate position sets in each decoding step, then the overall number of the decoding paths existing in the total $T$ decoding steps is $(2^N)^T = 2^{TN}$. With the constraint that all tokens in $Y$ should be predicted, we should omit the condition that there exist several tokens that are not be predicted during the whole decoding process from the total condition is $2^{TN}$. For the specific conditions that there are a number of $m$ tokens that are not be predicted, we should select the candidate tokens in the next $N - m$ tokens, then the number of this condition is $2^{T(N-m)}$, and we have $C_m^N$ to select these specific $m$ un-predicted tokens. We should consider the condition for each $m \in \{1, 2, ..., N\}$, and different $L_{\hat{Y}_i}$ have the repeat decoding paths. Actually, we can solve this problem with the Inclusion-Exclusion Principle (Andreescu et al., 2004). Thus, the total number of decoding paths is:*

$$2^{TN} - C_1^N 2^{(N-1)T} + C_2^N 2^{(N-2)T} - C_3^N 2^{(N-3)T} + ... = \sum_{m=0}^{N} (-1)^m C_m^N 2^{(N-m)T}.$$

## A.3 Details for the Search Times of Vanilla Path Selection Method.

**Lemma 2** *Given the predicted length $N$ of target sequence $Y$, the total decoding step $T$, the position beam number $k$, and the number of re-masked tokens in $t$th decoding step $n_t$, the total times for vanilla path selection method are $k * \sum_{t \in \{1,2,...,T\}} C_{n_t}^N$, and the search times for the simplified version are $T * k^2$.*

**Proof 2** *In $t$th decoding step, for each beam candidate, we select $n_t$ tokens from total $N$ tokens to be re-masked, thus the number of total candidates for single beam is $C_{n_t}^N$, and $k * C_{n_t}^N$ for total $k$ beams. Then, we should compute the total prediction probability for all $k * C_{n_t}^N$ candidates and select the highest $k$ ones for next decoding step. Thus the total search times for $T$ decoding steps are $k * \sum_{t \in \{1,2,...,T\}} C_{n_t}^N$. In the simplified version, we do not need to compute the total prediction probability for all $k * C_{n_t}^N$ candidates, we just replace one token to achieve the $k$ candidates for each single beam, and total $k^2$ for $k$ beams. Then we only need to compare the total prediction probability for these $k^2$ candidates and keep the highest $k$ ones, the search times are $k^2$, and $T * k^2$ for $T$ decoding steps.*

## A.4 Details of Generating the DPO Pairs

We present the details to generate the DPO pairs as mentioned in section 3.2 here. Given a specific training instance $(X, Y)$, where $Y$ is further decomposed into the mask parts $Y_{mask}$ and unmasked parts $Y_{obs}$, the reference model $\pi_{\text{ref}}$, we achieve the training pairs as follows:

(1) We enable $\pi_{\text{ref}}$ to sample the outputs of $Y_{mask}$, denoted as $O_{mask}$, where $O_{mask} = \pi_{\text{ref}}(Y_{mask}|Y_{obs})$, $\pi_{\text{ref}}(Y_{mask}|Y_{obs})$ denotes sampling the tokens in $Y_{mask}$ based on $Y_{obs}$, and the sampling method is to adopt the greedy output based on the prediction probability of $\pi_{\text{ref}}$.

(2) We randomly sample a subset of $O_{mask}$, denoted as $Y'_{mask}$, and and replace the tokens in $Y'_{mask}$ with the masked token, where the unmasked parts of $O_{mask}$ is denoted as $Y'_{obs}$.

(3) We sample the output of $Y'_{mask}$, denoted as $O'_{mask}$, where $O'_{mask} = \pi_{\text{ref}}(Y'_{mask}|Y'_{obs} \cup Y_{obs})$.

(4) We achieve one sampled output of $Y_{mask}$ as $Y'_{obs} \cup O'_{mask}$, denoted as $Y^1_{out}$.

(5) We repeat the above operation to achieve the other sampled output $Y^2_{out}$.

After obtaining the pair samples $Y^1_{out}$ and $Y^2_{out}$, we use a score function $\texttt{Score}(\cdot)$ to identify the positive and negative ones. Notice that we select the tokens with the highest prediction probabilities as the output when generating $O_{mask}$ and $O'_{mask}$, which is consistent with the Mask-Predict algorithm. Besides, we only sample the decoding path with two decoding steps to reduce the overhead during training, the different ratio to sample $Y'_{mask}$ from $O_{mask}$ has adapted the model to various masking conditions in different decoding steps during inference. In practice, we keep the ratio to sample the $Y'_{mask}$ the same during two sampling processes and determine it from a uniform distribution $U(0.2, 0.8)$. This is because once the ratio is large (e.g., 1.0), all tokens will be re-sampled again, and there is no difference between two sampling outputs, leading to meaningless pairs. Meanwhile, once the ratio is small (e.g., 0.01), only few tokens will be re-sampled again, there are many

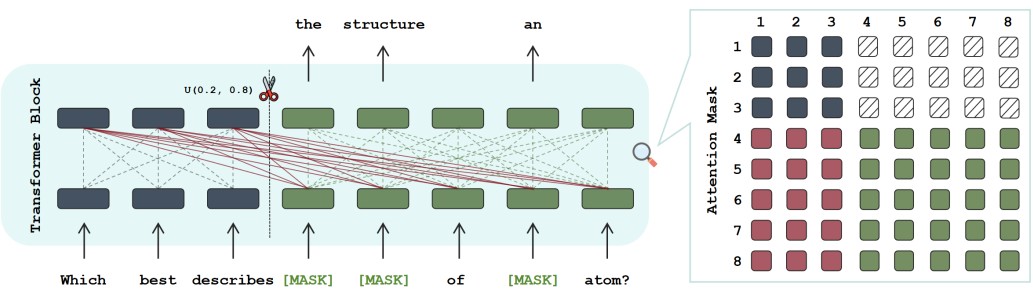

Figure 4: Presentation of generative masked language modeling.

overlaps between two $Y'_{obs}$, leading the sampling outputs $O'_{mask}$ lacking of diversity, which is not suitable for the DPO training.

## A.5 DETAILS FOR PRE-TRAINING TASK

We denote the pre-trained task of GeBERT as generative masked language modeling, which specially designed to fit the BERT-family to various generation tasks. This task is modified from the traditional masked language modeling (MLM) training objective, which makes the model learn to predict the specific masked tokens and has been widely used in traditional BERT-family models (Devlin et al., 2018; Liu et al., 2019). GeMLM aims to build a universal pre-trained BERT-family, which simultaneously possesses the ability of language understanding and generation. Motivated by the previous explorations in the NAR translation task (Ghazvininejad et al., 2019; Guo et al., 2020; Xiao et al., 2023) which extend the traditional MLM into the conditional generation scenery with the encoder-decoder model structure, and those that explore the potential in encoder-only models for language generation tasks (Wang & Cho, 2019; Liang et al., 2023b; Xiao et al., 2024), GeMLM first decomposes each training instance into two parts and assigns different masking strategies to help the model learn different capabilities. Besides, GeMLM further adopts the specific attention masking mechanism to enhance the consistency between the training and inference process.

Specifically, as shown in figure 4, given a specific training instance with the max context length $L$: $C = \{c_1, c_2, ..., c_{L-1}, c_L\}$, GeMLM decomposes $C$ into a tuple $(X, Y)$, where $X = \{c_1, c_2, ...c_{i-1}, c_i\}$ denotes the prefix tokens, and $Y = \{c_{i+1}, c_{i+2}, ...c_{L-1}, c_L\}$ denotes the suffix tokens. The prefix tokens are used to provide context information and help the model understand the whole sentence, we randomly sample a small ratio of mask tokens, which is similar to the traditional MLM in BERT, denoted as $(X_{mask}, X_{obs}) = \text{RANDOM\_MASK}(X, \beta_X)$, where $X_{mask}$ and $X_{obs}$ denote the masked and unmasked parts in $X$, $\beta_X$ denotes the masking ratio. The suffix tokens tend to help the model learn the generation capability, we adopt uniform masking as mentioned in CMLM (Ghazvininejad et al., 2019), denoted as $(Y_{mask}, Y_{obs}) = \text{UNIFORM\_MASK}(Y, \beta_Y)$, where $\beta_Y$ is sampled from a uniform distribution $U(0, 1)$. Then GeMLM predicts the masked tokens based on different context.

In practice, we adopt an adaptive masking function for the masking ratio $\beta_X$ as mentioned in Xiao et al. (2023) to replace the fixed masking ratio in the traditional MLM, as $\beta_X = 0.3 - \beta_Y * 0.2$. This operation can achieve more diverse masking conditions in $X$ for the model to learn and is based on the intuition that once more tokens in $Y$ are masked, $X$ should provide more context information (i.e., lower $\beta_X$). Besides, we prevent the query of each token in $X$ attending the tokens in $Y$ in the attention module as mentioned in figure 4 during training, which keeps consistent with the inference process since there is no target sequence in advance. Then, the final training loss of GeMLM can be computed as:

$$\mathcal{L}_{\text{GeMLM}} = - \sum_{x_t \in X_{mask}} \log \mathcal{P}(x_t | X_{obs}; \theta) \; - \sum_{y_t \in Y_{mask}} \log \mathcal{P}(y_t | X_{obs}, Y_{obs}; \theta). \quad (4)$$

## A.6 Details for pre-training

Details of the pre-training models and settings are present in Table A.6.

| Parameters | GeBERT-124M | GeBERT-352M |
|---|---|---|
| Num_layers | 12 | 24 |
| Hidden_size | 768 | 1024 |
| Num_attn_heads | 12 | 16 |
| Init_std | 0.02 | 0.02 |
| Seq_length | 2048 | 2048 |
| Batch_size | 1024 | 1024 |
| Train_iters | 153000 | 153000 |
| Learning_rate | 6e-4 | 3e-4 |
| Lr_decay_style | cosine | cosine |
| Clip_grad | 1.0 | 1.0 |
| Adam_beta | (0.9,0.95) | (0.9, 0.95) |
| Weight_decay | 1e-2 | 1e-2 |

Table 6: Details of the pre-training models and setting.

## A.7 Limitations and Broader Impacts

Our work demonstrates that BERT-family can perform better than AR language models by adopt our proposed path selection and path selection* methods. However, these models still require multi-step reasoning during zero-shot tasks to bridge the gap between inference and pre-training. This reasoning paradigm may affect the inference efficiency, making BERT-family models less effective than AR models in some contexts. On the other hand, this characteristic becomes an advantage in generative tasks. BERT-family with non-autoregressive generation paradigm do not need to perform sentence-length reasoning steps during decoding, leading to faster generation and reduced latency. This makes BERT-family particularly well-suited for tasks where quick text generation is critical, such as large-scale text generation.

