# OpenReview forum: "Path Selection Makes BERT-family Good Generators"
_ICLR.cc/2025/Conference — Submitted to ICLR 2025_

### Official Review · Reviewer_zyki · 2024-10-29

**Soundness:** 2
**Presentation:** 2
**Contribution:** 2
**Rating:** 3
**Confidence:** 4

**Summary:**

This paper examines BERT-family models' generation capabilities. It proposes path selection to expand inference search space and path selection*, which is an application of DPO, to train the model for better path selection. The methods improve BERT's performance across multiple zero-shot tasks to match or exceed autoregressive models.

**Strengths:**

This paper proposes innovative solutions - path selection to expand inference search space and path selection* to integrate path selection into training, which demonstrably elevate BERT's performance across multiple zero-shot tasks to match or exceed autoregressive models.

**Weaknesses:**

The generation capabilities of BERT are investigated in previous paper, e.g. [1]. This paper aims to improve the performance of BERT by using path selection, a non-autoregressive variation for inference. However, the paper lacks details on path selection, especially about how to choose the position to predict if beam size > 2. There are other potential issues should be considered for Mask-Predict methods, e.g. the potential mode collapse with many iterations. More ablation studies about the designs and clear pseudo-codes are needed.

Path selection* is an application of DPO, but the `Score' function is not defined in the paper, which diminishes the contribution of the paper.

[1] Patel, Ajay, et al. "Bidirectional Language Models Are Also Few-shot Learners." The Eleventh International Conference on Learning Representations.

**Questions:**

How to choose the position to predict if beam size > 2? In your case (beam size = 2) , the masked position with the highest probability is choose, but if beam size > 2, both topk and one-by-one replacement is possible. So the necessary details are lacked.

---

### Official Review · Reviewer_g4ao · 2024-11-03

**Soundness:** 2
**Presentation:** 1
**Contribution:** 2
**Rating:** 3
**Confidence:** 3

**Summary:**

The paper proposes a method to select the decoding paths of BERT-family to conduct generation tasks. They also propose a DPO method to optimize the preferences of the model over difference paths. Experimental results show that the decoding method and preference optimization method can effectively improve the generation quality of BERT-family on zero-shot commonsense reasoning, reading comprehension, summarization and question generation tasks.

**Strengths:**

1. The proposed method select the positions to decode dynamically, which is suitable for BERT-like models. BERT-like models trained with masked language modeling objective can predict any positions in the target part, while autoregressive language models can only follow the left-to-right order.
2. Experimental results show that the proposed method can effectively improve the generation quality of BERT-like models on various tasks.

**Weaknesses:**

1. On Line 146 the authors aim to identify an optimal decoding path. In other parts the authors also state that previous methods cannot find the optimal paths. But the proposed method cannot guarantee to find the optimal decoding path. It is a form of beam search over the mask-predict algorithm, and beam search cannot guarantee to find the optimal solution.
2. The description of the path selection algorithm in Section 3.1 is unclear. The example in Figure 2 is not enough since it didn't describe how the lowest-k total prediction probabilities are selected. The authors should give a pseudocode to describe the algorithm.
3. In Section 5.1 the authors find that BERT-like models can achieve competitive performance with AR models when the model predicts one token in each decoding step. In that case it requires the same number of decoding steps as AR models, and the compute of each decoding step is much larger than that of AR models since BERT-like models cannot reuse kv cache during inference. There is not efficiency advantage compared with AR models. Therefore I challenge the motivation to apply BERT-like models to generation tasks.
4. On Line 87 the authors claim that path selection* incorporates path selection into the training process. In fact neither the sampling methods to generate the pairs nor the computation of the probability considers the path selection method.

**Questions:**

1. What does it mean that "only one position in masked parts can be replaced by the one in unmasked parts to obtain the candidate decoding states"? Does it mean that only one position is predicted at each decoding step?
2. Is the algorithm equivalent to mask-predict when $k$ equals 1?
3. What is $n_{\text{new}}$ in Table 2? If it is equal to 1 as Table 1, how can the model achieves speeup over AR models?

---

### Official Review · Reviewer_ddFt · 2024-11-04

**Soundness:** 3
**Presentation:** 3
**Contribution:** 2
**Rating:** 6
**Confidence:** 3

**Summary:**

This work introduces GeBERT, a variant of BERT specifically pre-trained to leverage path selection, calming that it performs competitively, often on par with or surpassing traditional autoregressive models on various zero-shot and task-specific generation tasks.

**Strengths:**

1. With the rise of autoregressive models, research on BERT-family models for generative tasks has decreased, and few studies have explored their generative capabilities. This study takes a fresh approach to directly compare BERT models with AR models.

2. Introduces a method to expand the search space during inference, allowing BERT models to select the optimal path for improved generation quality. and also incorporates path selection into the training process, enabling BERT models to learn and prefer certain paths over others, further enhancing output quality.

3. Experimental results show substantial improvements in both zero-shot and fine-tuned settings, demonstrating that, with these modifications, BERT-family models can effectively compete with AR models.

**Weaknesses:**

1. While this research is innovative and valuable, a fundamental question remains—why use BERT for generation tasks? Given scaling laws, AR models generally improve significantly with larger model sizes, while BERT-family models struggle to achieve similar gains. Moreover, as shown in Table 1, the performance improvements for GeBERT are limited compared to AR models.

2. The path selection techniques introduce additional complexity, particularly in tuning hyperparameters for path selection*. This added complexity could hinder the practical application and usability of these methods.

3. The experimental models are relatively small in scale. Larger models might address or clarify the first weakness I raise, providing a stronger case for BERT's use in generation tasks.

**Questions:**

see above

---

### Official Review · Reviewer_pJpu · 2024-11-04

**Soundness:** 2
**Presentation:** 1
**Contribution:** 2
**Rating:** 3
**Confidence:** 4

**Summary:**

This paper investigates the application of BERT-style models as generators. It introduces two path selection strategies and trains a BERT-style model with instruction-following capabilities. The study demonstrates zero-shot performance in common sense question answering and reading comprehension, and achieves comparable autoregressive generation capabilities on the XSUM summary dataset.

**Strengths:**

-	In the context of the rising popularity of decoder-only architecture in large language models, this paper revisits the encoder-only architectures. This perspective may encourage the academic community to view generative models differently and inspire new thinking about model architecture.

-	The method proposed by the author endows BERT-style models with zero-shot capabilities in common sense question answering tasks, while also offering a speed advantage over autoregressive models.

**Weaknesses:**

-	While I believe research on encoder-only models remains valuable, the motivation behind this paper is unclear to me. Given that autoregressive models have demonstrated excellent generative capabilities across various tasks, why do the authors choose to focus on BERT-style models instead of enhancing autoregressive models? I look forward to discussing this with the authors.

-	If the authors aim to explore whether BERT-style models can achieve generative capabilities comparable to those of autoregressive models, then a broader range of generative tasks should be included in their experiments. The datasets used in the paper, such as ARC, primarily consist of multiple-choice questions and do not leverage generative capabilities.

-	If the authors intend to demonstrate that BERT-style models can achieve faster generation speeds, they should compare their speeds with a broader range of non-autoregressive models. However, it is unclear why the authors only use the encoder-decoder model BART as the baseline for this speed comparison in Table 2.

-	The experimental comparison does not seem to be fair. The author's pre-trained GeBERT uses advanced RoPE position encoding and sets the maximum length to 2048, but the maximum length of BART is 1024. More advanced generative models are recommended for comparison.

-	The writing needs improvement. The path selection algorithm is not clearly articulated, and there are also several typos present as follows:
a)	Section 3.1 discusses the path selection algorithm; however, the phrase “As shown on the left in Figure 2” in line 157 refers to the path selection-star illustrated in Figure 2.
b)	Line 372, “BART-largeGeBERT 124M” -> “BART-large”
c)	Line 535, “fron scratch” -> “from scratch”

**Questions:**

-	Why is the speed of Large Version not compared in Table 2?
-	In Table 1, how does GeBERT perform under fine-tuning as a traditional encoder-only model? This helps to judge the basic capabilities of GeBERT.

---

### Meta-Review · Area_Chair_1fdf · 2024-12-19

**Metareview:**

The paper argues that the way existing methods use BERT-like models for generation (what they are calling NAR-MLMs) is suboptimal. They claim that the random decomposition of the text sequence during training doesn't match the need to find a specific order during generation. The paper tackles a relatively under-explored area – using encoder-only models like BERT for generation. The experiments are criticized for their limited scope (lack of diversity in generative tasks, comparing against one AR model, lack of different sized models). The comparison to AR models isn't considered fair by some reviewers because of the experimental setup differences. The path selection algorithm isn't clearly explained, making it hard for reviewers to assess soundness. The reviewers ask for more analysis to better understand the role of the proposed methods and other possible issues in the algorithm.  Addressing the weaknesses, particularly improving clarity and providing more robust experiments, would be crucial for future submission consideration.

**Additional Comments On Reviewer Discussion:**

There is no author rebuttal.

---

### Decision · Program_Chairs · 2025-01-22

Reject